# Research on Multi-Dimensional Influencing Factors Regarding the Perceived Social Integration of New Urban Immigrants: An HLM Analysis Based on Data from 58 Large- and Medium-Sized Cities in China

**DOI:** 10.3390/ijerph19127017

**Published:** 2022-06-08

**Authors:** Fulian Li, Yang Sun, Youlu Zhang, Wuwei Zhang

**Affiliations:** 1School of Economics and Management, Shandong Agricultural University, Tai’an 271018, China; lifuliansdau@163.com; 2School of Public Administration, Shandong Agricultural University, Tai’an 271018, China; sunyang1422@126.com; 3School of Economics and Management, Beijing Jiaotong University, Beijing 100044, China; 18113024@bjtu.edu.cn

**Keywords:** new urban immigrants, perceived social integration, influencing factors, HLM model, China

## Abstract

Based on the China Migrants Dynamic Survey (CMDS) data from 2018 and the data from 58 large- and medium-sized cities in China, in this paper a hierarchical linear model was used to investigate the impact of demographic characteristics, social participation, and economic and social development on the perceived social integration of new urban immigrants at the individual and urban levels. The results revealed the following: (1) social participation, gender, age, education, health status, flow time and housing type of new urban immigrants had a significant positive impact on their perceived social integration, while income showed a U-shaped relationship with the sense of urban social integration; (2) macro-urban characteristics regulated the correlation between micro-individual factors and perceived social integration; (3) the significant advantages of new urban immigrants with higher education and more social participation in the process of integration into urban society were more obvious in cities with higher levels of economic development or public services. These findings enriched relevant research on the factors influencing the social integration of new urban immigrants and provided valuable insight with which the government could use to improve urban construction and promote the equalization of basic public services.

## 1. Introduction

With the accelerated development of urbanization and the market economy, the economic development level of central cities is constantly improving. The agglomeration capacity and the radiation capacity of resource elements have been gradually enhanced along with the increasingly obvious population agglomeration effect [1]. According to the “Seventh National Census Bulletin of China (No. 7)”, as of 2020, the migrant population in China reached 375.82 million, representing an increase of 69.73% since 2010. However, 124.83 million of these migrants are inter-provincial immigrants and the floating population within their provinces is 250.98 million. Since the reform and opening up of China, large-scale immigration has gradually become an important force to promote China’s economic growth [2]. At the same time, the ages of the migrants are decreasing. The size of the migrant groups in the post-1980s and post-1990s eras increased and their types of occupations were more extensive and diverse. A growing number of them were engaged in emerging occupations requiring high levels of knowledge and technical content [3,4,5]. Compared with the traditional old generation of “migrant workers”, the new urban immigrant groups, who have more advantages with respect to age and human capital, have become an important human resource for enhancing urban innovation and vitality and promoting urban economic development [6].

The social integration of new urban immigrants is a significant problem that cannot be ignored in the process of urban economic and social development. The better integration of new urban immigrants into urban society is not only an important embodiment of enhancing their sense of identity, but also the pursuit of the important value of promoting healthy and sustainable urban development and maintaining social harmony and stability. The existing literature shows that the most relevant studies on the influencing factors regarding the social integration of immigrants mainly involves the overall “migrants”, “migrant workers”, “ethnic minorities” and other specific groups, etc. Nevertheless, research on new urban immigrants is still lacking. As a significant symbol of China’s modern transformation, both in economy and society, new urban immigrants have their own characteristics in terms of lifestyle, career development, social relations, interest appeal, values and other aspects [7,8,9]. Therefore, the problem of their social integration needs to be focused on.

## 2. Literature Review

### 2.1. The Factors Regarding Influences on Social Integration

Many studies have revealed factors affecting the social integration of the migrant population. Currently, the academic community states that the differences in the social integration of migrants can be divided into two main aspects.

One aspect is represented by the microlevel factors of demographic characteristics. The characteristics of immigrants in terms of the individual, family and housing are important factors affecting their integration into urban society.

Individual characteristics, such as age, education level, income level, marital status, interpersonal communication and other variables, have a significant positive impact on the social integration of the floating population [10]. Based on the perspective of family-oriented migration, an excessive family-care burden would reduce the enthusiasm of the floating population for social integration [11]. However, family integrity in the process of migration is conducive to promoting the social integration and development of the floating population [12]. From the perspective of housing security, stable housing and good living environments could help to improve the stability of the lives of people in the floating population and speed up their integration into cities [13,14,15].

The other aspect is represented by external environmental factors at the macrolevel. Many studies have proved that the social integration of the floating population is inevitably affected by urban macro-environmental factors, this is due to the differences in the social integration of the floating population not only between different individuals, but also between different regions [16,17]. The higher the degree of inclusion in a city, the more conducive it is to the settlement and residence intention of the floating population and employment stability, thus promoting urban social integration [18,19,20]. However, megacities with strong urban characteristics will have a negative impact on the social integration of the floating population [21].

### 2.2. The Effect of Social Participation on Social Integration

For immigrants, social participation is embodied in consciously caring about public affairs and participating in the process of coconstruction, cogovernance and the sharing of urban social development [22]. The social support theory holds that persons who participate in the interactions of social networks can obtain access to emotional and instrumental resources [23,24]. On the one hand, they have more opportunities to obtain the support of social network relations through social participation, so as to help individuals to actively integrate into the urban society [25]; on the other hand, it is a valuable opportunity for the floating population to actively participate in the political, economic and cultural life of the community [26,27]. Increasingly, scholars have also argued for paying closer attention to the impact of social participation of different floating groups on social integration and analyses from the perspective of research methods [28,29,30]. Some scholars have also studied the social integration of the new generation of migrant workers [31], the elderly [32] and ethnic minority groups [33], and the results showed that social participation played a positive role in the social integration of different groups.

By reviewing the relevant literature, we found that the social integration of immigrants was not only affected by micro-individual factors, but also varied due to the urban characteristics at the macrolevel. For this reason, the essence of the social integration process of new urban immigrants was the result of the demographic characteristics, social participation and economic and social development.

In order to contribute to the literature, our study focused on the following questions: 1. how exactly is this process carried out? 2. what roles do the factors of different dimensions play in this transmission process? 3. are there any significant differences in the paths of social integration among new urban immigrants? For these problems, based on multi-dimensional data and from the subjective perspective of urban new immigrants, this article used the hierarchical linear regression model (HLM) to explore the influencing factors regarding the social integration of new urban immigrants and analyzed their hierarchical characteristics in depth. With the help of this article, we have shown that it is of great significance to promote urban new immigrants into the urban society sustainably, enhancing their sense of gain and happiness and give full play to the role of their human resources and, in the same way, to promote the harmonious and sustainable development of the economy and society and enhance the precision of government policies.

## 3. Materials and Methods

### 3.1. Data and Samples

This paper used microdata on 58 cities (or districts of municipalities directly under the central government) from the 2018 China Migrants Dynamic Survey (CMDS) for the empirical analysis. The macrodata were obtained from the 2019 statistical yearbooks of all provinces and cities. According to the concept definition of Jing et al. and Lian [34,35], the term new urban immigrants mainly refers to residents of the Chinese mainland area who were born after 1980 and are at least 16 years of age who work and live in the city and who have the willingness to stay in the city but have not obtained urban household registration. After obtaining the microdata, in order to avoid extreme and misleading samples, 56 cases (in which income was less than CNY 500 yuan or more than CNY 80,000 yuan) were deleted to make the data more uniform for the purpose of the research. We only kept the neighborhood committee samples in cities that had the following characteristics: a birth year after 1980; was at least 16 years of age; a “rural–urban” migrant direction; more than one year of migratory time; and a willingness to stay. Meanwhile, to fit into the analysis requirements of the HLM model, only cities with a sample size greater than 100 were selected, and the final effective sample size of the research study was 19,834, including data of 58 large- and medium-sized cities in China. Table 1 shows the sample size in the major cities. Meanwhile, as shown in Figure 1, the survey contained 58 cities (or districts of municipalities directly under the central government): Beijing, Shanghai, Chongqing, Shenzhen and Zhuhai in Guangzhou, Nanjing, Suzhou and Wuxi in Jiangsu, etc. The territories covered the eastern, central and western regions and represented different urban locations, population scales, economic levels and industrial types.

### 3.2. Hierarchical Linear Model (HLM)

Bryk and Raudenbush believe that hierarchical linear model (HLM) analysis methods are not limited by traditional regression statistical assumptions, such as independent residual errors, which make the estimation results of hierarchically structured data more accurate and stable [36]. The perceived social integration of new urban immigrants, which is not only affected by individual heterogeneity characteristics, but also varies due to the different economic and social development level of the cities, conforms to the characteristics of nested structure data; thus, the HLM model is suitable for analysis. The complete model equation expression is as follows:(1)$$Y_{ij}=\beta_{0j}+\beta_{1j}X_{ij}+\varepsilon_{ij}\;\beta_{0j}=\gamma_{00}+\gamma_{01}W_{j}+\mu_{0j}\;\beta_{1j}=\gamma_{10}+\gamma_{11}W_{j}+\mu_{1j}$$

The first equation is the individual-layer model, and the second and third equations are the city-layer models. Here, $Y_{ij}$ represents the dependent variable function of the *i*-th new immigrant individual in the *j*-th city, and $X_{ij}$ represents the observed value of the *i*-th new immigrant independent variable in the *j*-th city. $W_{j}$ represents the urban characteristic variables of the *j*-th city. In the individual-layer model, $\beta_{0j}$ and $\beta_{1j}$ indicate the intercept and slope of the dependent variable function of the *j*-th city in the independent variable regression line, respectively, and $\varepsilon_{ij}$ is the error term. In the city-layer model, $\gamma_{00}$ and $\gamma_{01}$ represent the intercept and slope of intercept $\beta_{0j}$ for the city feature variable in the regression lines, respectively, and $\mu_{0j}$ is the error term at the intercept of the individual-layer model brought by the city characteristic variables of the *j*-th city. $\gamma_{10}$ and $\gamma_{11}$ represent the intercept and slope of slope $\beta_{1j}$ for the city feature variable in the regression lines, respectively. $\mu_{1j}$ is the error term of the slope of the individual layer model brought by the city characteristic variables of the *j*-th city. At the same time, in order to reduce the multiple colinearity problem among the independent variables, improve the model stability and make the intercept term more realistic, the group mean centering method of independent variables in the individual and regional layers was used [37].

In view of the above, we constructed a theoretical model of the perceived social integration of new urban immigrants, as shown in Figure 2.

### 3.3. Variables

#### 3.3.1. Explained Variable

The dependent variable Y is the perceived social integration (integration). The perceived social integration can more truly reflect the subjective evaluation and psychological attitude of new urban immigrants in terms of their urban integration status (current work and life). A four-point scale was used to measure the new urban immigrants’ perceived social integration, in which one represents “strongly disagree” and four represents ”entirely agree”. There were 8 specific questions: ① “I like the city that I live in now”; ② “I pay attention to the changes in the city I live in now”; ③ “I am very willing to integrate into the local population and become one of them”; ④ “I feel like the locals are willing to accept me as one of them”; ⑤ “I think I am already a native”; ⑥ “I feel like locals look down on outsiders”; ⑦ “It is more important for me to follow the customs of my hometown”; ⑧ “My hygiene habits are quite different from those of my local citizens”. Then, to ensure the consistency of integration direction, the scores of questions ⑥, ⑦ and ⑧ were reverse transformed. After the dimension reduction principal component factor analysis with the help of SPSS 23.0 software (IBM, New York, NY, USA), only five questions, ①–⑤, were retained and summed up. The KMO spherical test value of 0.823 was significant at the 1% level; ultimately, this index was used as the measure of the perceived social integration of new urban immigrants.

#### 3.3.2. Explaining Variable

This article mainly explored the influence of variables at the levels of the individual and different cities on the evaluation of migrants’ subjective social integration. Independent variables X in the individual layer include the following. (1) Social participation (participation): concretely, the social participation of new urban immigrants manifested as the actual behaviors of community management, providing advice and suggestions, democratic supervision, volunteer activities and other social activities. The degree of social participation was obtained through four questions: ① “Do you give suggestions to your unit/community or supervise your unit/community management?”; ② “Do you report situations or make policy suggestions to the relevant government departments in various ways?“; ③ ”Do you comment online and participate in the discussion on state affairs and social events?“; ④ “Do you actively participate in donations, voluntary blood donation and volunteer activities?“. As for social integration, a four-point scale was used to measure the frequency of participation, in which one represents “never participate“ and four represents “often take part in these activities”. In the same way, all scoring items were summed up. (2) Gender (gender): dummy variable, where females = 0 and males = 1. (3) Age (age): continuous variable. (4) Marriage status (marriage): dummy variable, where single = 0 (including unmarried, widowed and cohabiting) and married = 0 (including first marriage and remarriage). (5) Degree of education (education): dummy variable, where primary school or below = 1, junior high school = 2, senior high school or technical secondary school = 3, junior college degree = 4 and bachelor’s degree or above = 5. (6) Health condition (health): dummy variable, where “can’t take care of themselves in life” = 1, “unhealthy but can take care of themselves” = 2, “basically healthy” = 3 and “perfectly healthy” = 4. (7) Income (income): in order to compress variable scales and avoid the impact of extreme data, we took the logarithm of new urban immigrants’ monthly income, simultaneously, and added the square of income (sq_income). (8) Migrant time (time): continuous variable. (9) Housing type (housing): classified according to the degree of personal ownership; dummy variable, where employer housing, borrowed housing, place of employment and other informal residence = 1, rented house = 2, self-bought and self-built houses = 3.

Independent variable X in the urban layer included the size of the urban permanent resident population (population; logarithm), per capita GDP (gdp; logarithm) and per capita public financial expenditure (finance; logarithm). Details of the variables are shown in Table 2.

## 4. Results

### 4.1. Construction and Results Analysis of the Null Model

The null model with no predictor variables at neither the individual level nor the city level mainly examined whether the individual perceived social integration varied significantly at the city level; its equation expressions are shown below.

Individual layer:(2)$$Y_{ij}=\beta_{0j}+\varepsilon_{ij}$$

City layer:(3)$$\beta_{0j}=\gamma_{00}+\mu_{0j}$$

The parameter estimation results are shown in Table 3. The within-group variance of the perceived new urban immigrants’ social integration was 5.142, and the inter-group variance value was 0.513. The chi-squared value was 1737.445, and both the fixed effect and the random effect were significant at the 1% level. The ICC was 0.091, somewhere between 0.059 and 0.138, which was considered significant by Cohen [38]. The above showed that the perceived social integration of new urban immigrant varied significantly among different cities, and the multi-layer linear model method was needed to further increase the predictive variables of the urban level to analyze the random effects.

### 4.2. Construction and Results Analysis of the Random Effects Model

In the individual-layer model, the observation variables such as social participation, individual characteristics and housing type of the new urban immigrant were included, and the variables in the urban layer model were randomized. The expressions are as shown below.

Individual layer:(4)$$\begin{array}{ll} Y_{ij}=\beta_{0j}+ & \beta_{1j}(participation)+\beta_{2j}(gender)+\beta_{3j}(age)+\beta_{4j}(marriage)\\ & +\beta_{5j}(education)+\beta_{6j}(health)+\beta_{7j}(income)+\beta_{8j}(sq\_income)\\ & +\beta_{9j}(time)+\beta_{10j}(housing)+\varepsilon_{ij} \end{array}$$

City layer:(5)$$\beta_{0j}=\gamma_{00}+\mu_{0j},\;\beta_{1j}=\gamma_{10}+\mu_{1j},\;\beta_{2j}=\gamma_{20}+\mu_{2j},\;\beta_{3j}=\gamma_{30}+\mu_{3j},\;\beta_{4j}=\gamma_{40}+\;\mu_{4j},\;\beta_{5j}=\gamma_{50}+\mu_{5j},\;\beta_{6j}=\gamma_{60}+\mu_{6j},\;\beta_{7j}=\gamma_{70}+\mu_{7j},\;\beta_{8j}=\gamma_{80}+\mu_{8j},\;\beta_{9j}=\;\gamma_{90}+\mu_{9j},\;\beta_{10j}=\gamma_{100}+\mu_{10j}$$

The results of the random effects model are shown in Table 4. The Chi-squared tests of the variance components of the random effects in the urban layer are shown. The intercept term test was significant at the 1% level, which showed that predictor variables at the city layer needed to be added to further explain their variation. Meanwhile, the random effect tests of participation, education, health, time and housing were also significant at different levels, indicating that the slope of these variables may have random effects and the marginal effect would be disturbed by the predictor variables in the urban layer.

### 4.3. Construction and Results Analysis of the Mixed Effects Model

Based on the results of the random effects model, we sequentially added predictor variables at the city level, such as population, gdp and finance, in the city-layer model. After the repeated screening of the effects of the independent random variables and interference effects of the urban layer variables, the expressions of the mixed effects model containing both individual-layer and city-layer variables was determined as shown below.

City layer:(6)$$\beta_{0j}=\gamma_{00}+\gamma_{01}\left(population\right)+\gamma_{01}\left(gdp\right)+\gamma_{01}(finance)+\mu_{0j},\;\beta_{1j}=\gamma_{10}+\;\gamma_{12}(finance)+\mu_{1j},\;\beta_{2j}=\gamma_{20},\beta_{3j}=\gamma_{30},\;\beta_{4j}=\gamma_{40},\;\beta_{5j}=\gamma_{50}+\gamma_{51}\;(gdp)+\gamma_{52}(finance)+\mu_{5j},\;\beta_{6j}=\gamma_{60}+\mu_{6j},\;\beta_{7j}=\gamma_{70},\;\beta_{8j}=\gamma_{80},\;\beta_{9j}=\;\gamma_{90}+\mu_{9j},\;\beta_{10j}=\gamma_{100}+\mu_{10j}$$

The results of the random effects model are shown in Table 5.

#### 4.3.1. The Influence Results of Individual-Level Predictor Variables

The key variable, social participation, had a significantly positive influence on urban social integration (the coefficient was 0.155, which was significant at the 1% level). The results showed that the social participation of new immigrants in urban community public affairs was beneficial to enhance their sense of social belonging and social integration.

The influence coefficient of gender on social integration was significantly positive (the coefficient was 0.101, which was significant at the 1% level), indicating that new female urban immigrants had a stronger sense of social integration.

There were significant positive correlations between age and migration time with perceived social integration (the coefficient values were 0.024 and 0.040, respectively, which were significant at the 1% level), which gradually increased with age and residence time in the city. Both education and health status had a significantly positive impact on immigrants’ sense of urban integration (the coefficient values were 0.138 and 0.555, respectively, which were significant at the 1% level). The higher the level of education, the higher the individual’s socioeconomic status and the healthier their physical condition was, the stronger their social and human capital.

There was a significant non-linear relationship between income and perceived social integration of new urban immigrants (the coefficient was −1.047, which was significant at the 10% level). With the increase in income, the sense of social integration of new urban immigrants decreased slowly, reaching a low point at about CNY 3146 yuan; then, the sense of social integration increased with the increase in income. There was a strong positive correlation between housing types and social integration (the coefficient value was 0.300, which was significant at the 1% level). Therefore, the stronger the sense of ownership of existing housing was, the stronger the sense of urban integration.

#### 4.3.2. The Influence Results of Urban-Level Predictor Variables

It could be concluded from the analysis of the impact of city-layer prediction variables on the perceived social integration of new urban immigrants (the coefficient value was −1.225, which was significant at the 1% level) that the factor related to GDP had a strong, negative impact on social integration, while fiscal expenditure had a significant positive effect (the coefficient value was 0.508, which was significant at the 5% level). These results showed that in more economically developed cities, due to household registration restrictions and human capital disadvantages, there was a larger gap between new urban immigrants and local urban residents in terms of living and working conditions, and thus a lower perceived social integration.

### 4.4. Robustness Check

In order to further ensure the reliability of the research conclusions and consider the lag of social development, this paper conducted a sample robustness test on the hierarchical linear models from the aspects of samples.

First, a sample robustness test was performed. Considering the lag of social and economic development, the samples of old members were removed. We replaced the macrolevel of the population size, fiscal expenditure per capita and GDP per capita with 2017 data. Next, the final complete model was analyzed again. The obtained result was still significant at the 1% significance level (see Table 6), which showed that the sample had good robustness.

## 5. Discussion

### 5.1. Main Findings

This study investigated the relationship between social participation and integration of immigrants into urban society. The results strongly proved that improving social participation played an important role in promoting the social integration of new urban immigrants.

The statistical results also supported the important relationship between economic and social development factors and social integration. Since economic income was directly related to housing satisfaction, career satisfaction and social satisfaction to a large extent, when these needs were basically satisfied in the society of the emigration place, migrants naturally had a higher evaluation of the overall situation of the emigration place.

As expected, demographic characteristics were crucial factors influencing the social integration of new urban immigrants. The degree of economic integration of new immigrants of different genders was significantly different. Compared with men, women had stronger demands for reunion and companionship, and, once settled down, they were more inclined to invest their income in real estate than men. In addition, the level of education had a positive impact on the identity integration of new urban immigrants, which was consistent with previous studies [39,40,41]. This was because people with more years of education were more likely to have a household registration and, hence, they suffered less social exclusion.

Moving time, housing type and educational background also had positive effects on the integration of new immigrants. This was due to the fact that the longer the migrants stayed in their destination, the more they accumulated relevant labor experience and other human capital and the more likely they were to achieve economic success and improve economic integration. The possibility of identity integration and cultural integration increased accordingly. At the same time, the increase in human capital with mobile time achieved the same effect as years of education, making the occupation more stable and allowed migrants to identify more with the local identity.

Compared with previous studies, scholars usually used multiple regression analyses to study the social integration of the floating population. However, such empirical estimation results might have a “one size fits all” hierarchical bias [42]. Therefore, this article used the HLM model to study the social integration factors of new urban immigrant groups, which could combine the individual characteristic variables of the floating population with regional variables. The combination of multiple models has high practical value.

### 5.2. Implications for Practice

Based on the results of this study, the following suggestions were put forward to substantially promote the sense of urban social integration of new urban immigrants in China.

First, it is recommended that we should strengthen group coordination and integrate social capital. On the basis of fully respecting and protecting the rights of new urban immigrants’ equal social participation, the government should achieve the following: form different groups in the process of communication considering heterogeneous interests that feel real; strengthen social interactions and emotional communication with local residents; effectively resolve social contradictions and psychological barriers; enhance awareness; engage new urban immigrants in order to generate the feeling of social integration.

Second, we suggest that the government should improve the urban construction and functions, accelerating the reform of the urban and rural household registration system and residence policy. While striving to further clarify the employment system, we should also improve the socialist-market economic system. A housing supply system should be established, whereby the government provides basic security, and the market satisfies different levels of demand so as to ensure that new urban immigrants can live in housing.

Third, the government should be pay attention to external empowerment and self-improvement, enhancing human capital, including the promotion of regional education resources and increasing basic education investment. These measures would effectively improve the quality of new migrants’ education and level. This will not only prove that substantive progress has been made in solving practical problems such as employment, income, housing, children’s education, social security and vocational skills training but also in promoting the development of new migrant families and society in urban areas.

### 5.3. Limitations and Future Research

Our study, however, might still have some limitations. Firstly, the subjective social integration of new urban immigrants is more likely to be a dynamic changing process. In terms of time changing trends, we only used cross-sectional data to analyze the current perceived social integration. It was difficult to achieve a synchronization of change. Accordingly, it was possible to adopt a longitudinal study design that could be more direct and efficient in future studies. Secondly, with the continuous development of China’s social economy, the factors influencing the decision making of new urban immigrants are likely to become more complex and diverse. We only focused on economic factors and public service factors from a macroperspective. Future studies could refine the impact of these two factors on the perceived social integration of new urban immigrants. In addition, new urban immigrants are the featured product of China’s rapid urbanization development and urban and rural household registration system, but the factors that affect global urban immigration decisions have common features, as reported by many scholars in Europe and America who have conducted in-depth research on social integration in populations [43,44]. Hopefully, our results can help further research on the social integration of global migrants. In future work, we will focus on the development of theories regarding urban immigrants in European and American regions, as well as other types of immigrants.

## 6. Conclusions

Based on the China Migrants Dynamic Survey (CMDS) data from 2018 and the economic and social development data of 58 large- and medium-sized cities in China, this paper established a hierarchical linear model to analyze the multidimensional influencing factors and the degree to which they affect the perceived social integration of new urban immigrants at the individual and city levels. Based on our studies and discussions, we drew conclusions as follows:

The perceived social integration of new urban immigrants, which was not only affected by individual factors but also varied due to the group characteristics at the city level, had a typical hierarchical characteristic;The perceived social integration of new urban immigrants, which was not only affected by individual factors but also varied due to the group characteristics at the city level, had a typical hierarchical characteristic;At the microlevel, many demographic characteristics of new urban immigrants played significant roles in improving their sense of urban social integration, such as social participation, gender, age, education, health status, migratory time and housing type;There was a U-shaped non-linear correlation between the income of new urban immigrants and their perceived social integration, and the inflection point was about CNY 3146 yuan;At the macrolevel, per capital GDP had a significantly negative impact on the subjective social integration of new urban immigrants, while per capital fiscal expenditure had a significant positive impact on social integration;At the cross-layer interaction level, per capital GDP strengthened the impact of educational background on perceived social integration, and fiscal expenditure had a significant strengthening effect on the impact of educational background and social participation on perceived social integration.

Moreover, although China is in the midst of large-scale urbanization, the phenomenon of anti-urbanization is gradually emerging. Many people migrate to cities, resulting in serious urban problems, including overcrowding, environmental pollution, resource shortages, etc. Therefore, according to the influencing factors of social integration, the question of how to improve the degree of social integration of new immigrants in new cities in China, so as to enable as many new immigrants as possible to build friendships with local city residents, beautify the urban environment and improve the happiness of the whole society, has attracted more and more attention from policymakers and scholars. Therefore, further research on related issues needs to be carried out.

## Figures and Tables

**Figure 1 ijerph-19-07017-f001:**
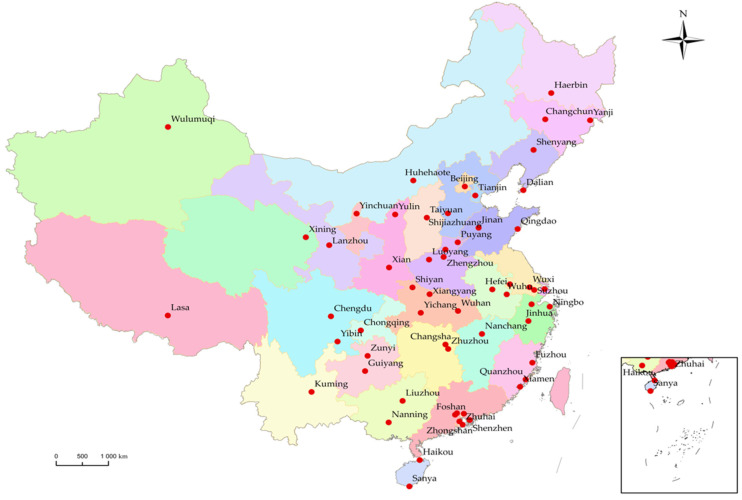
Location map of the sampled cities.

**Figure 2 ijerph-19-07017-f002:**
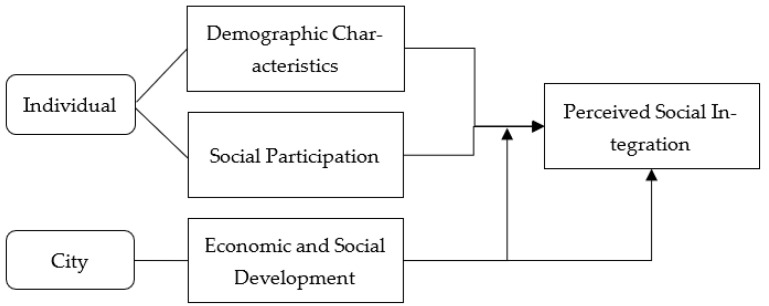
Model of perceived social integration of new urban immigrants.

**Table 1 ijerph-19-07017-t001:** Research samples from major cities (PPS method).

Province	City	Sample Size	Province	City	Sample Size
Beijing	Beijing	884	Shanghai	Shanghai	798
Chongqing	Chongqing	1007	Tianjin	Tianjin	831
Guangdong	Guangzhou	378	Jiangsu	Nanjing	563
	Shenzhen	673		Suzhou	392
	Zhuhai	100		Wuxi	430
Henan	Zhengzhou	639	Hubei	Wuhai	494
	Luoyang	148		Shiyan	167
	Puyang	122		Yichang	125
	Xinxiang	164		Xiangyang	123
Fujian	Fuzhou	102	Zhejiang	Hangzhou	382
	Quanzhou	179		Ningbo	139
	Xiamen	619		Jinhua	161
Anhui	Hefei	627	Sichuan	Chengdu	537
	Wuhu	137		Yibin	118
Liaoning	Shenyang	436	Hunan	Changsha	389
	Dalian	341		Zhuzhou	132

**Table 2 ijerph-19-07017-t002:** Characterization of variables.

Variables	Sample Size	Min	Max	Ave	Sd
Individual layer					
integration	19,834	5	20	16.39	2.36
participation	19,834	4	16	4.90	1.25
gender	19,834	0	1	0.47	0.50
age	19,834	16	38	30.37	4.72
marriage	19,834	0	1	0.74	0.44
education	19,834	1	5	2.81	1.03
health	19,834	1	3	2.88	0.33
income	19,834	6.21	11.29	8.29	0.56
sq_income	19,834	38.62	127.46	69.08	9.51
time	19,834	2	38	6.33	4.53
housing	19,834	1	3	2.12	0.62
City layer					
population	58	4.02	8.04	6.36	0.76
gdp	58	10.43	12.15	11.41	0.39
finance	58	8.85	10.90	9.60	0.44

**Table 3 ijerph-19-07017-t003:** Results of the null model on the influencing factors of the perceived social integration of new urban immigrants.

**Fixed Effect**	**Coefficient**	**SE**	**DF**	**T-Value**	***p*-Value**
Average integration level	16.435	0.095	57	172.397	0.000
**Random Effect**	**Variance**	**SD**	**DF**	**Χ^2^**	** *p* ** **-Value**
Regional-layer effect	0.513	0.716	57	1737.445	0.000
Individual-layer effect	5.142	2.268			

**Table 4 ijerph-19-07017-t004:** Results of the random effects model on the influencing factors of the perceived social integration new urban immigrants.

Variables	Fixed Effect		Random Effect	
	Coefficient	SE	Variance	SD
intercept	16.434 ***	0.095	0.515 ***	0.719
participation	0.157 ***	0.018	0.008 ***	0.091
gender	0.104 **	0.037	0.015	0.122
age	0.022 ***	0.005	0.000	0.019
marriage	−0.072	0.054	0.047	0.216
education	0.129 ***	0.026	0.021 ***	0.145
health	0.558 ***	0.059	0.053 *	0.230
income	−0.880 *	0.490	2.096	1.448
sq_income	0.056 *	0.059	0.007	0.082
time	0.038 ***	0.004	0.000 **	0.016
housing	0.281 ***	0.046	0.064 ***	0.254

Note: * *p* < 0.1, ** *p* < 0.05, *** *p* < 0.001.

**Table 5 ijerph-19-07017-t005:** Results of mixed effects model on the influencing factors of the perceived social integration new urban immigrants.

Variables	Fixed Effects		Random Effects	
	Coefficient	SE	Variance	SD
Individual layer				
intercept	16.437 ***	0.079	0.338 ***	0.581
participation	0.155 ***	0.018	0.006 ***	0.090
gender	0.101 **	0.037		
age	0.024 ***	0.005		
marriage	−0.110	0.055		
education	0.138 ***	0.026	0.018 ***	0.134
health	0.555 ***	0.058	0.053 **	0.244
income	−1.047 *	0.583		
sq_income	0.065 *	0.035		
time	0.040 ***	0.004	0.000 **	0.016
housing	0.300 ***	0.045	0.064 ***	0.244
City layer				
population	−0.069	0.108		
gdp	−1.225 ***	0.244		
finance	0.508 **	0.197		
Interactive layer				
education * gdp	0.140 **	0.059		
education * finance	0.117 **	0.050		
participation * finance	0.076 **	0.034		

Note: * *p* < 0.1, ** *p* < 0.05, *** *p* < 0.001.

**Table 6 ijerph-19-07017-t006:** Results of the mixed effects model on the influencing factors of the perceived social integration new urban immigrants (2017).

Variables	Fixed Effects		Random Effects	
	Coefficient	SE	Variance	SD
Individual layer				
intercept	16.436 ***	0.083	0.398 ***	0.631
participation	0.155 ***	0.018	0.009 ***	0.093
gender	0.101 **	0.037		
age	0.023 ***	0.005		
marriage	−0.110	0.055		
education	0.132 ***	0.026	0.021 ***	0.143
health	0.557 ***	0.059	0.058 **	0.242
income	−1.016 *	0.590		
sq_income	0.064 *	0.035		
time	0.040 ***	0.004	0.000 **	0.016
housing	0.298 ***	0.045	0.060 ***	0.245
City layer				
population	−0.110	0.120		
gdp	−1.321 **	0.227		
finance	0.293 **	0.197		
Interactive layer				
education * gdp	0.118 *	0.106		
education * finance	0.026 *	0.085		
participation * finance	0.011 **	0.049		

Note: * *p* < 0.1, ** *p* < 0.05, *** *p* < 0.001.

## Data Availability

The data used to support the findings of this study are available from the corresponding author upon request.

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
