# Peer review of "Research on Multi-Dimensional Influencing Factors Regarding the Perceived Social Integration of New Urban Immigrants: An HLM Analysis Based on Data from 58 Large- and Medium-Sized Cities in China"

_ijerph, 2022, doi:10.3390/ijerph19127017_

Round 1

Reviewer 1 Report

The manuscript deals with an interesting topic in the Chinese context, and could also be a potential suggestion of research subject to other territories/countries. I consider that this paper could be a good contribution to the topic. However, there are some recommendations for changes that are necessary. I suggest the following points to assist the author(s):

- Theoretical framework corresponds to the analysed subject, however some of the text in subsection 2.2 should be mentioned in the next section since is linked to the methodology.

- The name of section 3 could be changed, because it is not only about Data and Variables, as it includes the methods and materials, as well the procedures. My recommendations are a clearer organization of this section and a clearer presentation of the data processing.

- A review in the analysis/discussion of the results so that they may have more consistency. Another issue is that you do not mention this study’s limitations and you should provide them.

- The choice was made to merge the discussion and conclusion sections. It would be clearer if you presented these sections in separate so that the findings could be discussed more comprehensively, given the context of the study.

- Finally, a general English revision is advised as to simplify the understanding of the manuscript, as well as enhance its readability. Whilst proofreading, please correct some spelling mistakes and consider rephrasing some sections. I have added a few cases below, just as examples, to help you in this process:

Line 2 and 3 (Title): Research on the Muliti (?), Immingrants

Line 36: were inter-provincial immigrant

Line 38: large-scale immigrant ( immigrant or immigration?)

Line 43: consider rephrasing to avoid repetition of the word “more”

Line 49: is extremely a problem (needs grammar correction)

Line 55: the researches (similar case on line 67-68)

I believe it is worth re-working your manuscript to show the value of this research.

Author Response

General response:

Thanks for your recognition of our work and giving us the opportunity to revise and resubmit a revised version of our manuscript Research on the Multi-dimensional Influencing Factors of the Perceived Social Integration of New Urban Immigrants: A HLM Analysis Based on Data from 58 Large and Medium-sized Cities in China” (ijerph-1705234). We appreciated the constructive criticisms of the reviewers. The manuscript has certainly benefited from these insightful revision suggestions. All the authors have seriously discussed about all these comments, we have tried our best to modify our manuscript to meet with the requirements of "International Jouranl of Enviornomental Research and Public Health". In this revised version, changes to our manuscript within the document were all highlighted by using “Track Changes”. As you are concerned, there are some problems that need to be addressed. According to your nice suggestion, we have made necessary corrections to our previous draft, the detailed corrections are listed below.

1. Theoretical framework corresponds to the analysed subject, however some of the text in subsection 2.2 should be mentioned in the next section since is linked to the methodology.

Response:

      We sincerely appreciate your valuable comments. There are indeed related methods are introduced in the text 2.2, we consider putting the hot spots and shortcomings of previous literature, research framework and significance in the next section (3.Materials and Methods), Moreover, we have adjusted and tried our best to complete the subsection 2.2 in our revised manuscript. Your constructive remark indeed makes our study more convincing.

2. The name of section 3 could be changed, because it is not only about Data and Variables, as it includes the methods and materials, as well the procedures. My recommendations are a clearer organization of this section and a clearer presentation of the data processing.

Response:

       Thank you for your comments. Your proposal is very important to improve the content of our manuscript. We have re-name the section 3-----Materials and Methods. Furthermore, Because the part of molding belongs to measurement method and counting process of the paper,we choose to adjust it as part of the section 3.3.According to the adjusted content,we divide the section 3 into three parts as follow:

3.1. Data and Sample

3.2. Hierachical Linear Model (HLM)

3.3. Variables

3.3.1. Explained Variable

3.3.2. Explaining Variable

      We sincerely appreciate the valuable comments. The revised supplementary WB has been uploaded. Changes to our manuscript within the document were all highlighted by “change track”. We believe these modifications will make our work more scientific

3. A review in the analysis/discussion of the results so that they may have more consistency. Another issue is that you do not mention this study’s limitations and you should provide them.

Response:

      Thanks for your comments. In order to make the analysis/discussion more accurate, we modified the original presentation in the “Conclusion” section. We really benefit from your comments because they make our manuscript easier to understand.

      As you pointed out, our study may still have some limitations. Firstly, the subjective social integration of new urban immigrants is more likely to be a dynamic changing process, in terms of time changing trends, we only use cross-sectional data to analyze the current perceived social integration, it is difficult to achieve synchronization of change. Accordingly, it is possible that adopt a longitudinal study design can be more directly and effciently in future studies.

      Secondly, with the continuous development of China's social economy, the factors influencing the decision-making of new urban immigrants are likely to become more complex and diverse. We only focused on economic factors and public service factors from a macro perspective. Future studies could refine the impact of these two factors on the perceived social integration of new urban immigrants.

      In addition, new urban immigrants are the featured product of China's rapid urbanization development and urban and rural household registration system, but the factors that affecting global urban immigration decisions have common features. Hopefully, our results can help further research on the social integration of global migrants. In the next research direction, we are now also working on the development of theories on urban immigrants in European and American regions as well as other types of immigrants

4.The choice was made to merge the discussion and conclusion sections. It would be clearer if you presented these sections in separate so that the findings could be discussed more comprehensively, given the context of the study

Response:

      We appreciate your suggestions. After reading a large number of relevant literature, many articles have separate discussion sections and conculsion sections. We have adjusted the structure of the article presenting each of these two parts as separate chapters, so that readers would understand the content of the article more clearly and organized.

      Furthermore, in the discussion section, we modified and improved the content appropriately. More imprtantly, we put frward detailed suggestions to promote the degree of social integration between new immigrants and local immigrants from different subjects, so that the content of the article is more targeted and applicable. Changes to our manuscript within the document were all highlighted by “change track”. We believe that your constructive remarks will substantially improve our articles.

5. Finally, a general English revision is advised as to simplify the understanding of the manuscript, as well as enhance its readability. Whilst proofreading, please correct some spelling mistakes and consider rephrasing some sections. I have added a few cases below, just as examples, to help you in this process.

Response:

      We sincerely appreciate the valuable comments. We have corrected the spelling errors, which really make our manuscript more accurate. In addition, we have choosed MDPI-reccommended editing service to unfergo ectensibed English revisions.Spelling errors that have been modified in the revised manuscript have been corrected. we belibeve the revised version will make it easier for readers to understand our studies. The revised supplementary WB has been uploaded. We believe these modifications will make our work more scientific.

Thank you again for your positive comments on our manuscript. International Jouranl of Enviornomental Research and Public Health is an influential journal. From all the papers published in your journal, readers have been learning a lot. We sincerely hope that our article will be published in your journal. If there are any other modifications we could make, we would like very much to modify them and we really appreciate your help. Thank you very much for your help.

Reviewer 2 Report

1. Based on the social integration of new urban immigrants, the article analyzes data from personal and social factors such as intercept, partition, gender, population, finance, etc., and fully summarizes the main conditions for analyzing urban immigrants. It helps to draw comprehensive and reliable conclusions. And the data comes from the official Migrants Dynamic Survey, which can reflect the correct population flow information. The hierarchical linear models (HLM) analysis model is not limited by traditional regression statistical assumptions, and the hierarchical structure data is more accurate and stable. However, the data source is too single, and there is no control group as verification, and the reliability is questionable. At the same time, the single model only performs hierarchical processing on the data and does not consider the correlation between the data. Often these factors are related to each other in pairs and need to be further explored.
2. The literature review statement is too confusing and does not highlight the research focus, which is easy for readers to question, and the expression of Figure 1. Model of perceived social integration of urban new immigrants is not rigorous enough, and the author is expected to refer to other scholars' reference drawings.
3. The logic of the chapter setting structure is slightly unreasonable. For example, the conclusion and discussion are mixed together, making it difficult for readers to understand the specific content of the article.
4. From what perspectives do the 8 selected questions judge the sense of social integration of new urban immigrants? Why choose urban permanent resident population, per capita GDP, per capita public financial expenditure as independent variables? Please give reasons or justifications.
5. The chapter setting is a bit unreasonable. The data and model should belong to the research design, and the research results should be described separately.
6. Whether the cities selected in this study and the research results can be analyzed with maps or other visualizations can make the effect more intuitive.
7. Is the HLM method supported by relevant literature? The plausibility and reliability of the model used for this study cannot be determined at this time.
8. What is the internal relationship between Null Model, Random Effects Model and Mixed Effects Model? Why not reflect it in the abstract?

Author Response

General response:

Thanks for your recognition of our work and giving us the opportunity to revise and resubmit a revised version of our manuscript Research on the Multi-dimensional Influencing Factors of the Perceived Social Integration of New Urban Immigrants: A HLM Analysis Based on Data from 58 Large and Medium-sized Cities in China” (ijerph-1705234). We appreciated the constructive criticisms of the reviewers. The manuscript has certainly benefited from these insightful revision suggestions. All the authors have seriously discussed about all these comments, we have tried our best to modify our manuscript to meet with the requirements of "International Jouranl of Enviornomental Research and Public Health".Thank you for reading our manuscript and reviewing it, which did help us improve it to a better scientific level. In this revised version, changes to our manuscript within the document were all highlighted by using “Track Changes”. Point-by-point responses to you are listed as follows.

1.Based on the social integration of new urban immigrants, the article analyzes data from personal and social factors such as intercept, partition, gender, population, finance, etc., and fully summarizes the main conditions for analyzing urban immigrants. It helps to draw comprehensive and reliable conclusions. And the data comes from the official Migrants Dynamic Survey, which can reflect the correct population flow information. The hierarchical linear models (HLM) analysis model is not limited by traditional regression statistical assumptions, and the hierarchical structure data is more accurate and stable. However, the data source is too single, and there is no control group as verification, and the reliability is questionable. At the same time, the single model only performs hierarchical processing on the data and does not consider the correlation between the data. Often these factors are related to each other in pairs and need to be further explored.

Response:

      Thank you for your valuable comments. Based on a genenral linear regression analysis model, the hierarchical linear models (HLM) analysis model acutally is which can process data at both individual and population levels [1]. Acording to previous studies, the hierarchical linear models (HLM) analysis model has more advantages than the traditional linear regression model in terms of the conrol group [2]. Although the control group is not set separately, Li et al. confirmed that the hierarchical linear models (HLM) analysis model takes into accout the gruoup aggregation fator (i.e. the correlation and independence within the group are considered) [3].

      In Stephen et al.’s studies, this method which can effectively solve the problem of organization effect (or background effect) [4]. However, if the study does not consider the problem of “inter-group aggregation” at all, it is very reliable for verfication to set up control groups. The purpose of using random effect model and the hierarchical linear models (HLM) analysis model is totake into account not only individual effects but also group-level effects. At the same time, we also hope that in the future, there will be more reliable methods to further study the relationship between influencing factors, which is also one of the directions of our next research. Thanks again for your comments

References:

[1] Yong H. The Application of Multilevel Analysis Model in Online Collaboration Learning. Open Education Research.2011,17(01):80-85.

[2] Raudenbush, S. W., Bryk, S., Cheong, Y. F., & Congdon, R. HLM6: Hierarchical linear and nonlinear modeling. Chicago: Scientific Software International.2004: 66-91.

[3] Xue Y. L, Tao X. Hierarchical Linear Model for Binary Data: Principle and Application. Psychological Development and Education. 2006, (04):97-102.

[4] StephenW. Raudenbush, AnthonyS.Bryk. The hierarchical linear models (HLM): applications and data analysis methods. Social Literature Publishing House, 2007.

2.The literature review statement is too confusing and does not highlight the research focus, which is easy for readers to question, and the expression of Figure 1. Model of perceived social integration of new urban immigrants is not rigorous enough, and the author is expected to refer to other scholars' reference drawings.

Response:

      Thank you for your valuable comments. According to your suggestions, we have revised the literature review part of this arcitle, which includes two parts: the factors of the influence on social integration and the effect of social participation on social integration. Compared with the previous contents, we made the following major changes:

      (1). We tried to made previous literatures were summaried so as to make the literature review more refined and logical.

     (2). We tried to make a broader analysis of the connections and differences between China and other countries in the global social integration.

     (3). We tried to make a horizontal comparison in chronological order and summarize the literature so that readers can have a clearer understanding of the focus of the study.

     (4). We tried to make a longitudinal description of the historical evolution, development status and trend prediction of social integration and social participation

      In additon,we have referred to the reference drawings of other scholars, Combined with the research content of this article, we have redesigned the model of perceived social integration of new urban immigrants. The new figure has been uploaded to the revised manuscript. Changes to our manuscript within the document were all highlighted by “change track” . In the process of revision, we deepen our understanding of this field. Your constructive remark indeed makes our study more convincing.    

3.The logic of the chapter setting structure is slightly unreasonable. For example, the conclusion and discussion are mixed together, making it difficult for readers to understand the specific content of the article.

Response:

      Thanks for your nice suggestion. Your proposal is really important to improve our work. We have adjusted the structure of our article according to your suggestions:The content of “discussion” is the fifth part of this article, and the content of ”conclusion” is the sixth part of this article. At the same time, we also put forward several specific suggestions from different levels in the revised “discussion” section to improve the degree of social integration between new urban immigrants and local residents. In order to improve the scientific nature of our studies, we also have listed the limitations of this paper in the revised “conclusion” section. We believe the revised version will make it easier for readers to understand our studies.

4.From what perspectives do the 8 selected questions judge the sense of social integration of new urban immigrants? Why choose urban permanent resident population, per capita GDP,per capita public financial expenditure as independent variables? Please give reasons or justifications.

Response:

      We thank you for rasing those questions. Here is an an explanation of the logical connection between the 8 questions chosen to measure the sence of social integration of new urban immigrants. Social integration is a muli-dimensional progressive concept [1], different scholars divide social intergration into different dimensions, it includes economic, cultural, and psychological dimensions [2-3]. Since some scholars defined that social integration can only be truly achieved if individuals have a sense of identity and belonging to the place where they have migrated [4-5]. It is now widely accepted that psychological integration as a process of change in terms of psychological and emotional identification with one’s own social identity and belonging, as well as the social distance between oneself and local residents [6]. With the above facts, in our study, we try to judge the sence of social integration from different perspective. Then we chosen the qusestion of (1)(3)(4)(5)(6), these questions were all about the perspective of psychological integration. Besides, the question of (7) and (8) were about the perspective of acculturation, the question of (2) was about the perspective of economic integration.

      As for the chice of independent variables, in Wu’s studies, according to modern push-pull theory, floating population can choose the degree of social integration efforts by balancing the "push" and "pull" between urban and rural areas [7]. This makes the current urban resident population one of the facts to judge the degree of social integration of floating population [8], which providing beneficial inspiration for this study. At the same time, per capita GDP and per capita public financial expenditure is a major indicator of the economic dimension of social integration. Many studies have shown that cities in relatively developed areas are places of concentrated migration. This is mainly due to higher per capita income, more employment opportunities, and in greater fiscal contribution these regions [9]. Base on the above evidence, we choose urban permanent resident population, per capita GDP, per capita public financial expenditure as independent variables.

References:

[1] Yang, J. Segregation, Selective Assimilation and Assimilation: A Conceptual Framework of Rural to Urban Migrant’s adaptation at destination. Popul. Res. 2009, 33, 17–29.

[2] Zhang, W.; Lei, K. The new urban immigrants’ social inclusion: Internal structure, present situation and influential factors. Sociol.Stud. 2008, 137, 117–141.

[3] Yang, J. Citizenization of migrants: Theory, reality and reflection. Jilin Univ. J. Soc. Sci. Ed. 2019, 33, 100-110.

[4] Yang, J. Research on the assimilation of the flfloating population in China. Soc. Sci. China 2015, 230, 61-79.

[5] Li, T.C.; Chu, C.C.; Meng, F.C.; Li, Q.; Mo, D.; Li, B.; Tsai, S.-B. Will happiness improve the psychological integration of migrant workers? Int. J. Environ. Res. Public Health. 2018, 15, 900.

[6] Yue, Z.S.; Li, S.Z.; Feldman, M.W. Concept construction and empirical analysis of social integration for rural-urban migrants in China. Mod. Econ. Sci. 2012, 34, 1-11.

[7] Wu, Y.F.; Lei, X.K.; Nie, J.L.From Structure to Cognition:Social Capitial and Social Integration of Floating Population—Based on the CLDS Survey Data in 2014.Population and Development. 2019,25(5):111-112.

[8] Zhu, Y.; Lin, L.Y. Mobility patterns of floating population and their social protection: Moving from’ Uraban Inclusion’ to ’Social Inclusion’ ScientiaGeographica Sinica.2011,31(3):264-271.

[9] Zhang, Y.; Cen, Q. Spatial patterns of population mobility and determinants of inter-provincial migration in China. Popul. Res.2014, 38, 54-71.

5.The chapter setting is a bit unreasonable. The data and model should belong to the research design, and the research results should be described separately.

Response:

      We sincerely appreciate the valuable comments. We have readjusted the structure of the article in the revised manuscript. We have re-name the section 3——Materials and Methods. In addition, the data samples and urban locations are visualized.The structure of the section 3 ——Materials and Methods is shown below:

3.1. Data and Sample

3.2. Hierachical Linear Model (HLM)

3.3. Variables

3.3.1. Explained Variable

3.3.2. Explaining Variable

      At the same time, we separately have described the research results of the article and adjusted the contents of this part into the section of 4.3.1 and 4.3.2. In the revised manuscript, we have summarized the research resules and path cofficients from in dividual and urban level predictor variables. The social integration path of new urban immigrants was shown in Figure 4. Changes to our manuscript within the document were all highlighted by “change track”. We believe that your suggestions are very helpful to the improvement of our manuscript.

6.Whether the cities selected in this study and the research results can be analyzed with maps or other visualizations can make the effect more intuitive.

Response:

      Thank you for your comments. Detailed explanations of the cities selected for our studies and the results of the studies are carried out have been included in the article. The survey contained 58 cities (or districts of municipalities directly under the central government), Through screening,the total number of effective samples in 58 cities was 19834, which basically avoided the problem of uniformity of sample.

      In the revised version, 58 cities have been marked with red dots on the map in Figure 1, the number of samples in each city is also have been presented in Table 1. As shown in Figure 4, we have marked the explanatory degree of our studies results which were also analyzed visually as much as possible. We believe the revised version will make it intuitive for readers to understand our studies.

7.Is the HLM method supported by relevant literature? The plausibility and reliability of the model used for this study cannot be determined at this time.

Response:

      We appreciate your suggestions. Hierarchical Linear Model (HLM) is able to exert analyzing hierarchical data with the characteristics of nested structures. Many aticles have reprted the application of Hierarchical Linear Model (HLM) in the study of social integration of floating population. Xue used Hierarchical Linear Model (HLM) to investigat the factors and their effect all the social integration of floating population [1]. In Guo et al.’s studies, Hierarchical Linear Model (HLM) also was used to examine floating population's social integration and the results of the social integration measured by intentions to stay at destination cities [2]. According to Su et al., Hierarchical Linear Model (HLM) was applied to analyze from two levels of farmers and rural areas [3]. In Li’s studies, based on the national floating population data and public service equalization data, Hierarchical Linear Model (HLM) was used to analyze the influencing factors of China's floating population happiness [4]

      Due to the aim of our studies is to investigate the impact of demographic characteristics, social participation and economic and social development on the perceived social integration of new urban immigrants from the perspectives of individual layer and urban level. Base on the above evidence, our studies involved data at the individual and population levels. As a result, we used Hierarchical Linear Model (HLM).

References:

[1] Xue Y.A Research on the Factors Affecting Social Integration of Floating Poputation Based on Hierarchical Linear Model.2016, (03):62-72.

[2] Guo M., Fu.ZH. Ethnic Minorities in an Age of Mobility:Status and Consequences of Social Integration in a Comparative Perspective——An Empirical Study Based on Data from the CMDS 2017. Qinghai Journal of Ethnology.2020,31(03):53-70.

[3] Su T., Yang HJ., Xian XJ., Cao XL. Research on Influencing Factors of Farmers’ in Suburban Villages Based on HLM Model——The Case Study of 1441 Households in the Greater Xi’An Area. Chinese Journal of Agricultural Resources and Regional Planning.2020,41(04):273-282

[4] Li Q. Analysis on the influencing factors of happiness of floating population. Statistics & Decision.2015(21):92-95.

8.What is the internal relationship between Null Model, Random Effects Model and Mixed Effects Model? Why not reflect it in the abstract?

Response:

      Thank you for your valuable comments. In our study, Null Model was the first step of HLM model analysis without adding any independent variables. It was used to examine whether there are significant inter-layer differences in dependent variables, and then to judge whether multi-layer analysis is necessary. If significant differences exist, multiple levels of analysis are necessary.

      Random Effects Model was the second step of HLM model analysis which only added one level independent variable, so we incuded the social participation, individual characteristics, housing type and other observational variables of new urban immigrants. Only the first layer has predictive variables and the second layer has none in this model.

       Mixed Effects Model was the last step of HLM model analysis which includes the independent variables of the city level and the individual level. Through the comparative analysis of the Random Effects model and Mixed Effects model, we could clearly see the change of the intercept slope and significance of the model after the addition of rural variables. Thanks again for your comments.

      Thank you again for your positive comments on our manuscript. International Jouranl of Enviornomental Research and Public Health is an influential journal. From all the papers published in your journal, readers have been learning a lot. We sincerely hope that our article will be published in your journal. If there are any other modifications we could make, we would like very much to modify them and we really appreciate your help. Thank you very much for your help.

Reviewer 3 Report

The major problem of the work is the theoretical conception of the concept of social integration. It is perceived as a final and static state, when social integration is a dynamic process. Moreover, social integration is also a complex process in which both the arriving migrants and the host population are part of. And in this work it seems that social integration only depends on the characteristics of the arriving group, as if the receiving context has no influence, beyond the characteristics of the different cities.

So, as the work is presented, the concept of social integration should be replaced by others that better define the situation, such as accommodation.

Its measurement of social integration lacks key elements such as the origin (not only foreign) of immigrants (urban/rural), generation, or cultural patterns (religion, language, etc.) and places of origin.

Nor does it develop Chiswick's theory ("U") either theoretically or conceptually, and it would require further development since it is fundamental to the results.

The dependent variable (social integration) as it is presented is of perception (subjective) and not of objective indicators that show if there is a difference between what is perceived and the objective indicators of how they are accommodated.

We have no information on which cities the immigrants arrived in. Nor the number of participants.

In the discussion there is no reference to what happens in other places: Europe or America.

Make a review of the broader discussion.

Author Response

General response:

Thanks for your recognition of our work and giving us the opportunity to revise and resubmit a revised version of our manuscript Research on the Multi-dimensional Influencing Factors of the Perceived Social Integration of New Urban Immigrants: A HLM Analysis Based on Data from 58 Large and Medium-sized Cities in China” (ijerph-1705234). We appreciated the constructive criticisms of the reviewers. The manuscript has certainly benefited from these insightful revision suggestions. All the authors have seriously discussed about all these comments, we have tried our best to modify our manuscript to meet with the requirements of "International Jouranl of Enviornomental Research and Public Health". In this revised version, changes to our manuscript within the document were all highlighted by using “Track Changes”. Point-by-point responses to you are listed as follows.

1.The major problem of the work is the theoretical conception of the concept of social integration. It is perceived as a final and static state, when social integration is a dynamic process. Moreover, social integration is also a complex process in which both the arriving migrants and the host population are part of. And in this work it seems that social integration only depends on the characteristics of the arriving group, as if the receiving context has no influence, beyond the characteristics of the different cities.

So, as the work is presented, the concept of social integration should be replaced by others that better define the situation, such as accommodation.

Response:

      Thank you for your valuable time and advice. Here is some clarification about the internal reasons for the selection of the scope of social integration in our study. In the study of the concept of social integration, the method of constructing multi-dimensional indicators to measure the degree of integration has been widely adopted [1-2]. In our study, we regarded social integration as a dynamic progressive multidimensional and interactive process, so the social integration was seen as a process of economic and cultural adaptation and psychological identity formed by the interaction between the floating population and the destination subjectively. "new immigrants" was a cross-concept with "floating population", which referred to the recognition of all people who enter the place without official change of household registration, and corresponded to the local population [3].

      The reason why the local population is not connected in this article was that in order to avoid the institutional barrier of the place of migration. Besides, in Zhou’s studies showed that the relationship between the various dimensions of the inflow and outflow places of floating population is also controversial [4].so considering the availability of data, our evaluation of the degree of integration mainly stays in the study of the influx group. If possible, we will try to explore the multi-dimensional evaluation factors of the combination of inflow and outflow.

References:

[1] Yang JH. Index of assimilation for rural to urban migrants: A further analysis of the conceptual framework of assimilation theory. Population and amp; Economics.2010(2):64-70.

[2] Kang L, Zhang WZ, Chen L. Multidimensional measurement and analysis of social integration of low-income communities in Beijing. Hum a Geography.2019,34(3):22-29.

[3] Tong X, Ma XH. "Harmonizer" and "Integration" -- community integration of new immigrants in cities. Social Science Research.2008(1):77-83.

[4] Zhou H. Measurement and theoretical perspectives of immigrant assimilation in China. Population Reasearch. 2012,36(3):27-37.

2.Its measurement of social integration lacks key elements such as the origin (not only foreign) of immigrants (urban/rural), generation, or cultural patterns (religion, language, etc.) and places of origin.

Response:

      Thanks for your nice suggestion. Several studies have reported the factors about of social integration, the social integration of floating population exists in all countries of the world. As for the social integration of international immigrants, western studies mainly have focused on the relationship between immigrants and mainstream society [1]. In Cordon’s study, Cultural contact, ethnic identity, prejudice and discrimination were used to measure social integration [2]; Park’s study showed that generational integration and assimilation are regarded as "a process of mutual infiltration and integration of groups and individuals [3].

      However, different countries have different situations of new urban migrants and different factors to measure social integration.However, in China, the migration of urban and rural population among the new urban immigrants is the mainstream of migration. This social integration is mainly reflected in the economic, cultural and psychological aspects [4]. Based on the above evidence and our studies, our study explored the influencing factors of social integration from a subjective perspective

References:

[1] Li MH. Theory of migration in the 20th century. Journal of Xiamen University (Arts & Sciencea). 2000(04):12-18+140.

[2] Cordon M.M. Assimilation in American Life. New York: Oxford University Press.1964.

[3] Park R.E. Community Organization and the Romantic Tempe.Robert E Park & Ernest.1974.

[4] Chen Y, Wang J. Social integration of new-generation migrants in Shanghai China.Habitat International. 2015,49:419-425.

3.Nor does it develop Chiswick's theory ("U") either theoretically or conceptually, and it would require further development since it is fundamental to the results.

Response:

      Thank you for your valuable comments. Your proposal is really important to improve our work. As you said, the results of our study acutally were the basis for the development of the theory of Chiswick. Compared with the theoretical and conceptual development, our study actually has focused on the application of Chiswick’s theory in practice, which was also the main purpose of our study. Our study has took the role orientation of new immigrants as the starting point of the study. As a new field of study on labor migration, we have considered the new urban immigrants as the suppliers of labor force and the investors of human capital. We believed that this is a better application of Chiswick’s theory in China.We believe that your suggestions are very helpful to the improvement of our manuscript.

4.The dependent variable (social integration) as it is presented is of perception (subjective) and not of objective indicators that show if there is a difference between what is perceived and the objective indicators of how they are accommodated.

Response:

      We sincerely appreciate your valuable comments. As you said, social integration was actually divided into objective and subjective aspects, the former mainly refers to the explicit objective social integration such as economic integration and behavioral adaptation, while the latter mainly refers to the implicit subjective social integration such as value concept identity [1-2]. Existing researches and policies mostly have focused on the comprehensive or objective aspects of subjective integration [3], however, a few studies on subjective integration mainly have focus on individual dimensions such as identity, sense of belonging and subjective well-being [4-5].

      Therefore, the purpose of our study is to explore the influencing factors of the subjective social integration of new urban immigrants from the perspective of the feelings and experience of the floating population. In terms of data availability, subjective indicators do reflect social integration better than objective indicators, which is what we are concerned about.

References:

[1] Goldlust JR, Anthony H. A Multivariate Model of Immigrant Adaptation <sup/>. International Migration Review. 1974.8(2), 193–225.

[2] Yang J. Index of assimilation for rural to urban migrants: A futuher analysis of the conceptual framework of assimilation theory. Population and amp; economics.2010(2):64-70.

[3] Yue ZS, Du HF, Li SZ. Social integration: Definitions, theories, and its applications. Journal of Public Management.2009,6(2):14-121.

[4] Wu YF, Lei XK, Nie JL. From structure to cognition: Social capital and sovial integration of floting population-based on the CLDS survey data in 2014.Population and Development. 2019,25(5):112-122.

[5] Moody J, White DR. Structural cohesion and embeddedness: A hierarchical concept of social group. American Sociological Rreview.2003,68(1):103-127.    

5.We have no information on which cities the immigrants arrived in. Nor the number of participants.

Response:

       We appreciate your comments. In fact, our studies contained fifty-eight cities (or districts of municipalities directly under the centreal government): Beijing, Shanghai, Chongqing, Shenzhen and Zhuhai in Guangzhou, Nanjing, Suzhou and Wuxi in Jiangsu and so on, covering the eastern, central and western regions and representing different urban locations, population scales, economic levels, and industrial types. At the same time, we have put 58 cities on the map as red dots of Figure 1 in the revised version so that which make our manuscript more intuitive.

      The micro data of our studies used the 2018 Chinese Migrants Dynamic Survey (CMDS) data, with a total sample number of 169989. After effective screening, the final sample number is 19834. The sample size in the major cities have been represented of Table 1 in the revised version. By this way, we sincencerely hope to make it easier for readers to see the data in this article.

6.In the discussion there is no reference to what happens in other places: Europe or America.

Response:

      Thank you very much for your valuable comments and information. We have refined the discussion section in the revised version. We believe these modifications will make our work more scientific. In the discussion section of the revision, compared the research results with those of other countries, we have carried out an extensive discussion and proposed the limitations of this paper and several suggestions for improving social integration. We hope that this will make our writing more general.

7.Make a review of the broader discussion.

Response:

      We are so grateful for your kind suggestion. We have readjusted the discussion section according to your suggestion, the discussion section have been divided into three parts, including: Main Findings (5.1) , Implications for Practice (5.2) and Limitations and Future Reseaech (5.3).

      In Main Findings (5.1), we have summarized the factors that influence social integration degree and analyzed the causes of this result in detail. At the same time, the innovation of this article has been also presented in this part.

      In Implications for Practice (5.2), based on the results of this study, we have put forward three effective suggestions on promoting the social integration degree of new urban immigrants from different angles.

      In Limitations and Future Reseaech (5.3), as for the limitations of the article, we also have summarized the limitations of this article from three perspectives form reality. In addition, we predict and expand the hot spots of this study which is also our future research dirertion.

      Thank you again for your positive comments on our manuscript, which did help us improve it to a better scientific level. International  Jouranl of Enviornomental Research and Public Health is an influential journal. From all the papers published in your journal, readers have been learning a lot.We sincerely hope that our article will be published in your journal. If there are any other modifications we could make, we would like very much to modify them and we really appreciate your help. Thank you very much for your help.

Round 2

Reviewer 1 Report

The authors have considerably revised the manuscript and responded to all recommendations.

The argument of the manuscript and the structure of the sections are more coherent, thus better organized and have added value to the paper, as well as greater readability.

Author Response

Dear editors:

Thanks for your recognition of our work and giving us the opportunity to revise and resubmit a revised version of our manuscript Research on the Multi-dimensional Influencing Factors of the Perceived Social Integration of New Urban Immigrants: A HLM Analysis Based on Data from 58 Large and Medium-sized Cities in China” (ijerph-1705234). We feel grateful for your professional review work on our article. The manuscript has certainly benefited from these insightful suggestions. All the authors have seriously discussed about all these comments, we have tried our best to modify our manuscript to meet with the requirements of " International Jouranl of Enviornomental Research and Public Health". According to your nice suggestion, we have made necessary corrections to our previous draft. In this revised version, changes to our manuscript within the document were all highlighted by using “Track Changes”.

1.The argument of the manuscript and the structure of the sections are more coherent, thus better organized and have added value to the paper, as well as greater readability.

Response: We are so grateful for your kind suggestion. We have checked manuscript carefully and the spelling errors, which really make our manuscript more accurate. Spelling errors that have been modified in the revised manuscript have been corrected. we belibeve the revised version will make it easier for readers to understand our studies. The revised supplementary MS Mord has been uploaded. We believe these modifications will make our work more scientific.

Thank you again for your positive comments on our manuscript. If you have any question about this paper, please don’t hesitate to inform us. We are pleased to do anything more that is helpful to improvement of our manuscript.

                                                                  Your sincerely,

                                                                  Wuwei Zhang